# High-Density Chitosan Induces a Biochemical and Molecular Response in *Coffea arabica* during Infection with *Hemileia vastatrix*

**DOI:** 10.3390/ijms242216165

**Published:** 2023-11-10

**Authors:** Julio César López-Velázquez, Soledad García-Morales, Gloria Paola López-Sánchez, Mayra Itzcalotzin Montero-Cortés, Alberto Uc-Várguez, Joaquín Alejandro Qui-Zapata

**Affiliations:** 1Biotecnología Vegetal, Centro de Investigación y Asistencia en Tecnología y Diseño del Estado de Jalisco A.C., Camino Arenero 1227, El Bajío, Zapopan 45019, Mexico; jucelopez_al@ciatej.edu.mx; 2Biotecnología Vegetal, CONAHCYT-Centro de Investigación y Asistencia en Tecnología y Diseño del Estado de Jalisco A.C., Camino Arenero 1227, El Bajío, Zapopan 45019, Mexico; smorales@ciatej.mx; 3Escuela de Ingeniería y Ciencias, Tecnológico de Monterrey, General Ramón Corona 2514, Nuevo México, Zapopan 45201, Mexico; paola.lsibt@gmail.com; 4Instituto Tecnológico de Tlajomulco/TecNM, Km. 10 Carretera Tlajomulco-San Miguel Cuyutlán, Tlajomulco de Zúñiga 45640, Mexico; mayra.mc@tlajomulco.tecnm.mx; 5Subsede Sureste, Centro de Investigación y Asistencia en Tecnología y Diseño del Estado de Jalisco A.C., Tablaje Catastral 31264 Km 5.5 Carretera Sierra Papacal-Chuburna, Mérida 97302, Mexico; auc@ciatej.mx

**Keywords:** *Hemileia vastatrix*, *Coffea arabica*, chitosan, NPR1, coffee rust

## Abstract

The coffee industry faces coffee leaf rust caused by *Hemileia vastratix*, which is considered the most devastating disease of the crop, as it reduces the photosynthetic rate and limits productivity. The use of plant resistance inducers, such as chitosan, is an alternative for the control of the disease by inducing the synthesis of phytoalexins, as well as the activation of resistance genes. Previously, the effect of chitosan from different sources and physicochemical properties was studied; however, its mechanisms of action have not been fully elucidated. In this work, the ability of food-grade high-density chitosan (0.01% and 0.05%) to control the infection caused by the pathogen was evaluated. Subsequently, the effect of high-density chitosan (0.05%) on the induction of pathogenesis-related gene expression (*GLUC*, *POX*, *PAL*, *NPR1*, and *CAT*), the enzymatic activity of pathogenesis-related proteins (GLUC, POX, SOD, PPO, and APX), and phytoalexin production were evaluated. The results showed that 0.05% chitosan increased the activity and gene expression of *ß-1,3 glucanases* and induced a differentiated response in enzymes related to the antioxidant system of plants. In addition, a correlation was observed between the activities of polyphenol oxidase and the production of phytoalexin, which allowed an effective defense response in coffee plants.

## 1. Introduction

Coffee is classified as one of the agricultural products with the highest economic value worldwide [1]. In Mexico, coffee growing is an important economic activity, positioning it as the eleventh largest producer country in the world, with an estimated production of 900,215 tons in 2020 [2]. However, there are increasing threats to coffee production due to pests, diseases, and adverse climatic conditions. Among these limitations in coffee production, the most devastating disease is coffee leaf rust caused by *Hemileia vastatrix*. Under favorable conditions for the pathogen, it can cause defoliation of up to 50% and losses between 30 and 50% in production yield [3]. Resistant varieties and the application of agrochemicals, mainly copper-based, are used to control the disease [4]. Despite the release of some resistant cultivars in recent years, the rust continues to negatively affect production and undermine the income of producers due to the appearance of new strains and new outbreaks of the disease [5]. The use of agrochemicals continues to be the most widely used option for its control, although their use has several drawbacks, including their cost, toxicity, harmful effect on the environment, and the development of resistance, which makes their use difficult in urban areas and traditional low-scale crops. This makes it necessary and a priority to search for ecological and safe alternatives in the agricultural sector [6]. Among these alternatives, the use of elicitors has been explored, whose purpose is to activate the defense response of plants with the application of low concentrations and are characterized by being of natural or synthetic origin [7]; in coffee plants, the use of these molecules has been documented with positive effects [8,9,10].

Recently, research has been reported on the role of chitosan as a plant defense inducer in *Coffea arabica* for the control of coffee rust [11]. The mechanism of action of chitosan as an elicitor has been described for some plants, although not completely in coffee. Chitosan is recognized by plasma membrane receptors and induces the transcription of defense-related genes, which is reflected in the synthesis of proteins and metabolites of interest. It has been documented that chitosan increases transcriptional levels of β-1,3 glucanases, chitinases, PR1, Pti5, and WRKY26, phenylalanine ammonium lyase (PAL), resveratrol synthase, as well as the antioxidant enzyme system. In addition, it regulates stomata opening and stimulates the generation of reactive oxygen species (ROS) and energy production [12,13]. However, there are different chitosans on the market that vary in price, purity, source, and process of obtaining them, which leads to different physicochemical properties, and the response in plants can also vary.

In a study conducted by López-Velázquez et al. [11], the effect of the physicochemical properties of commercial chitosans of low and medium molecular weight, as well as practical (Sigma-Aldrich) and high density (food grade) chitosan was used. The results of this work showed that the viscosity of chitosan is an important parameter due to adherence to the plant, whereas molecular weight and degree of acetylation did not show differential effects. In addition, it was observed that chitosan influences the plant defense system, increasing the activity levels of β-1,3 glucanases and peroxidases. The biological effectiveness of coffee rust of chitosans from different methods of production, such as conventional chemical (practical grade) and biotechnological (enzymatic process), has also been evaluated [14]. The best protection response was observed with the chitosan obtained by the chemical method, presenting a higher level of PR enzyme activity (β-1,3 glucanases and peroxidases) in plants treated with chitosan and the pathogen. Both practical-grade chitosan and high-density chitosan showed higher protection against coffee rust [11]. However, practical grade chitosan has been extensively studied and is characterized by higher purity but also higher cost, which could discourage its use. However, high-density chitosan, although of lower purity, can be purchased at a lower price and is readily available, facilitating its use in agriculture.

The defense response induced by practical-grade chitosan has already been reported for coffee [11], although the biochemical and molecular response of this induced resistance remains to be described in more detail. In addition, it is necessary to verify whether the effect of defense induction in other chitosans is maintained or equivalent to that induced by practical grade chitosan. In this work, the biological effectiveness of two concentrations of high-density (food-grade) chitosan as an elicitor in the protection against coffee rust was evaluated. The concentration that induced the highest protection against rust was selected to identify the biochemically and molecularly induced defense mechanism triggered in coffee plants susceptible to *H. vastatrix*.

## 2. Results

### 2.1. Biological Effectiveness Test

The disease occurred in all treatments inoculated with *H. vastatrix*. However, the severity of the infection was different for each treatment. Plants in the control treatment (C) showed no signs of the disease, no defoliation occurred during the experiment, and no colonization of the pathogen was verified by microscopy (Figure 1). The treatment inoculated with the pathogen (H) showed 100% disease incidence, and the area under the disease progress curve (AUDPC) value was 102 (Table 1). In all cases where the disease was observed, the plants showed the characteristic symptoms of coffee rust, defoliation, leaves with chlorosis, and the appearance of yellow or orange spots on the leaves (Figure 1a,b). On the other hand, in the treatments that were inoculated with the pathogen and treated with high-density chitosan 0.01% (Q1) and 0.05% (Q5), although they presented 100% disease incidence, the severity of infection was lower in treatment Q5 (Table 1). Microscopy tests showed that the mycelium of the pathogen was present in both treatments, although, in the Q5 treatment, cell death was also observed in the infected tissue (Figure 1c).

### 2.2. PR-Protein Activity and Phytoalexin Accumulation

For the evaluation of the defense mechanisms induced by chitosan, a concentration of 0.05% was chosen. The activity of enzymes related to pathogenesis (PR) and plant defense, such as β-1,3 glucanases, peroxidases, superoxide dismutase, ascorbate peroxidase, and polyphenol oxidase, was evaluated, as well as the accumulation of phytoalexins through the content of total phenolic compounds. For β-1,3-glucanase activity, it was observed that plants inoculated with the pathogen (H) presented a significant decrease in activity starting at 12 h after inoculation with the pathogen that was maintained until 24 h (Figure 2a). An increase in the level of the control was observed at 48 h and a significant decrease at 72 h.

For plants inoculated with the pathogen and treated with chitosan (QH), a significantly differentiated response to infected plants was observed; at 12 h, a significant increase in activity was observed, decreasing to the level of the control at 24 h. After 48 h, a significant increase in activity was observed and then a decrease to the level of the control at 72 h. The response induced by chitosan in coffee plants for β-1,3-glucanase activity was always higher than that present in infected plants (H) (Figure 2a).

Regarding the group of enzymes with peroxidase activity, plants inoculated with the pathogen (H) did not show a significant difference with the control at times evaluated (Figure 2b). For plants inoculated with the pathogen and treated with chitosan (QH), a significantly different response to infected plants was only observed up to 48 h after inoculation with the pathogen. A decrease in activity was observed at 48 h and returned to the level of the control at 72 h (Figure 2b).

For ascorbate peroxidase activity, it was observed that plants inoculated with the pathogen (H) did not show a significant difference with the control at the times evaluated (Figure 2c). For plants inoculated with the pathogen and treated with chitosan (QH), only a significantly different response to infected plants was observed up to 72 h after inoculation with the pathogen, with a decrease in activity (Figure 2c).

On the other hand, for superoxide dismutase activity, it was observed that plants inoculated with the pathogen (H) presented a significant decrease in activity at 48 h after pathogen inoculation, although there was an increase in activity to the level of the control at 72 h (Figure 2d). For plants inoculated with the pathogen and treated with chitosan (QH), a significantly different response to infected plants was observed; at 24 h, a significant decrease in activity was observed, which increased to the level of the control at 48 h. At 72 h, a significant decrease in activity was again observed (Figure 2d).

Regarding the activity of the polyphenol oxidase enzyme (Figure 2e), it was observed that plants inoculated with the pathogen (H) showed a significant decrease in activity at 24 h after pathogen inoculation, although there was an increase in activity at the pathogen level at 48 h and a significant increase at 72 h (Figure 2e). For plants inoculated with the pathogen and treated with chitosan (QH), a significantly differentiated response to infected plants was observed; at 12 h, a significant decrease in activity was observed, which increased to the level of the control at 24 h and was maintained at 48 h. At 72 h, a significant increase in activity was again observed at the level of the pathogen (Figure 2e). At 72 h, a significant increase in activity was again observed (Figure 2d). The chitosan-induced response in coffee plants for polyphenol oxidase activity was early and higher than that observed in infected plants (H) (Figure 2e).

Finally, the accumulation of phytoalexins was evaluated with the quantification of total phenolic compounds, which are related to the capacity of the plant to stop infection against pathogens. It was observed that plants inoculated with the pathogen (H) showed a significant decrease from 12 h after pathogen inoculation, although there was an increase in activity at the pathogen level at 24 h. A significant decrease was observed at 48 h and again an increase to the level of the control at 72 h (Figure 2f). For the plants inoculated with the pathogen and treated with chitosan (QH), there were no significant differences with the control at 12 and 24 h. It was not until 48 h that the content decreased, although to a lesser extent than in infected plants, and an increase in the level of the control was observed at 72 h (Figure 2d). The response induced by chitosan in coffee plants for the content of total phenolic compounds was greater than that observed in infected plants (H) (Figure 2f).

### 2.3. Quantitative Analysis of Defense-Related Gene Expression by Real-Time PCR

Normalization of the reference genes used showed the following values: GADPH: 0.1726; actin: 0.1000, 14.3.3, and 0.1219, and the actin gene was chosen as the internal control gene. Gene expression of genes encoding proteins related to plant pathogenesis and protection was also determined. In the relative expression of the β-1,3-glucanase gene (Figure 3a), it was observed that plants inoculated with the pathogen (H) presented a significant increase in expression 12 h after inoculation with the pathogen and subsequently a significant decrease in its expression at 24 h that was maintained at 48 h. For plants inoculated with the pathogen and treated with chitosan (QH), a lower level of expression was observed than in infected plants at 12 h. However, after 24 h, a significant increase in the level of expression was observed, which decreased significantly, although in a greater proportion than the other treatments up to 48 h, and was maintained up to 48 h.

Peroxidase gene expression (Figure 3b) did not show significant differences with respect to the control at the times evaluated in plants inoculated with the pathogen (H). For plants inoculated with the pathogen and treated with chitosan (QH), a high level of expression was observed with respect to infected plants from 12 h, although it decreased at 24 h (seven-fold) and reached the same level of expression as the control at 48 h.

Regarding catalase gene expression (Figure 3c), it was observed that plants inoculated with the pathogen (H) showed a significant increase at 12 h after pathogen inoculation. Expression levels decreased at 24 h and remained at the level of the control at 48 h. For plants inoculated with the pathogen and treated with chitosan (QH), a level of expression was observed at the level of the control at 12 h, increasing at 24 h and a higher level of expression at 48 h with respect to infected plants (Figure 3c).

Likewise, the expression coding for NPR1 was evaluated (Figure 3d), and it was observed that plants inoculated with the pathogen (H) presented a significant increase at 12 h after pathogen inoculation (15-fold). However, expression levels decreased significantly to the level of the control at 24 h and remained at this level at 48 h. For plants inoculated with the pathogen and treated with chitosan (QH), a level of expression was observed at the level of the control at 12 h, increasing at 24 h (12-fold) and a higher level of expression at 48 h (52-fold) with respect to infected plants (Figure 3d).

Finally, in the evaluation of phenylalanine ammonium lyase (PAL) expression (Figure 3e), it was observed that plants inoculated with the pathogen (H) showed a significant increase at 12 h (40-fold) after pathogen inoculation. However, expression levels decreased significantly to the level of the control at 24 h and remained at this level at 48 h. For plants inoculated with the pathogen and treated with chitosan (QH), expression levels were observed at the control level at 12 h, increasing at 24 h (4-fold) and returning to the control level at 48 h (Figure 3d).

## 3. Discussion

There is a great diversity of chitosans on the market, with different origins and methods of obtaining them, from conventional chemical processes for the deacetylation of chitin to enzymatic and green transformation processes. This diversity has been found to confer differentiated properties, and in the case of its effect as a plant defense inducer, it may be greater or lesser depending on its physicochemical characteristics [11] or method of obtaining [14]. However, an important point to consider is that, depending on its nature and purity, the costs of chitosan can be variable. In this work, we chose to use high-density food-grade chitosan that has a lower cost with respect to other chitosans, which could facilitate its adoption as a control product for coffee rust, with a lower impact on the cost of production. In this study, it was observed that the food-grade chitosan maintained its control effect for coffee rust and plant defense induction, equivalent to that previously described with a practical-grade chitosan, the most widely used and described.

The effectiveness of chitosan in the present study is evident by demonstrating that the amount of disease estimated in terms of severity was lower for treatments Q1 and Q5, compared to the control treatment with an amount of disease of 25, 40, and 102, respectively. In addition, a notorious vegetative development proves the protective ability of chitosan to induce the immune system of plants as an effective agent in activating plant defense [15]. Chitosan has been shown to stimulate plant growth and defense hormones, such as jasmonic acid and salicylic acid, at low concentrations [16]. In this work, it was also possible to observe the induction of cell death in leaves infected with *H. vastatrix* at the highest concentration of chitosan (Q5).

There are similar studies in *Coffea arabica*, with Soares-Leal et al. [17] using copper and silver nanoparticles. In turn, Fajardo-Franco et al. [18] used biofungicides against yellow rust and Plaza-Pérez et al. [19] applied boron, zinc, and manganese for the same purpose.

In this work, the changes related to the induction of plant defense at the biochemical and molecular level were investigated. The function of a plant defense inducer was facilitated through a phenomenon called priming, where molecules are used that act as response modifiers that can lead to a more intense, faster, earlier, or more sensitive defense response compared to plants that are exposed to the same stress condition, as mentioned by Tugizimana et al. [20].

During the induction of the effective defense response, specialized defense mechanisms are triggered that have the function of preventing the establishment of the pathogen in the plant tissue. Pathogenesis-related proteins (PRs) are overexpressed under stress conditions or following pathogen attack. These responses are controlled by two signaling pathways, jasmonic acid (JA) and salicylic acid (SA), which have specific protein groups; for JA, one can cite plant defensin, thionin, and chitinase B. For the SA pathway, it is NPR1 which is considered the central regulator of defense responses in a large number of plants and involves both systemic acquired resistance (SAR) and induced systemic resistance (ISR) [15]. Chitosan is recognized to activate the transcription of the gene encoding *NPR1*. In a study carried out by Gangireddygari et al. [21], it was found that gene expression levels by foliar application of chitosan on pepper increased, and, in turn, upregulated genes such as *PAL*, *polyphenol oxidase* (*PPO*), and *superoxide dismutase* (*SOD*), keeping infection caused by a virus (CMV) limited. These results correlate with the results obtained in this work, where an increase in the transcriptional levels of *NPR1* was observed. With the application of chitosan (HQ), higher levels of expression were observed compared to other treatments at 48 and 72 h. It is worth mentioning that the NPR1 protein is found at basal levels in the nucleus and cytoplasm [12]. Prior to pathogen infection, the SA concentration is low, and NPR1 is inactivated through disulfide bonds. However, upon infection, the SA content increases and NPR1 switches to an active state and is transferred to the nucleus to regulate gene expression [12].

On the other hand, when a fungal infection occurs in plants and when chitosan is applied, genes related to the defense response are activated, among which are the enzymes β-1,3 glucanases and chitinases of the GH19 family (classes I, II, IV, VI, and VII) [14]. A correlation was observed with the results obtained between gene expression and enzyme activity of β-1,3 glucanases, where an increase in enzyme activities was observed at 12, 48, and 72 h in plants treated with chitosan and inoculated with *H. vastatrix* (QH); this response was observed more than 100 times in comparison with infected plants (H). For the gene expression assay, it was observed that chitosan had an increase from 12 h and remained elevated until 48 h, when it reached its maximum expression. In turn, this behavior is similar to that observed by Jogaiah et al. [15], who observed that cucumber plants increased the activity of these enzymes from 6 to 48 h after being inoculated with *Erysiphe cichoracearuma*. Additionally, Liu et al. [22] reported that chitosan induced *glucanase* gene expression in potatoes against *Alternaria tenuissima* and maintained their high levels compared to their controls. It is worth mentioning that β-1,3 glucanases are considered the best characterized pathogenesis-related proteins, which could directly hydrolyze the cell walls of phytopathogenic fungi, thus releasing β-1,3 glucan compounds and chitin oligosaccharides that stimulate host defense responses [23]. Therefore, it was suggested that the induction of this enzyme activity had an effect during the invasion of the pathogen in coffee plants since an increase in the activity of these enzymes and a decrease in the severity of the infection were observed, which would indicate that the enzymes were able to hydrolyze cellular parts of the pathogen [24].

During the pathogenesis process of different fungi, the production and accumulation of molecules that have different functions in the response process of the plant to the pathogen attack are triggered. Among these molecules, ROS have different functions in the plant defense response process, and they can act as second messengers that generate a response locally, including cell wall strengthening, protein cross-linking, and the hypersensitive response (HR), which can include the induction of programmed cell death (PCD) [25]. However, these molecules can also damage proteins and lipids and cause alterations in host DNA, necessitating the induction of enzymatic and non-enzymatic control mechanisms to prevent this damage [26]. Within all this cellular machinery are peroxidases which are enzymes that play a role in a wide range of physiological and developmental processes; they are involved in the generation and detoxification of hydrogen peroxide (H_2_O_2_), being antioxidants, and they are an important initial defense adapted by plants to cope with the effects of biotic and abiotic stresses [27]. The variation in the level of peroxidase activity is related to the type of elicitor, as not all resistance inducers increase the level of peroxidases compared to other levels of protection. It was reported by Fischer [28] that in some cases, peroxidase activity levels remained low or null, a behavior like that obtained in this study, where it was observed that *peroxidase* gene expression levels initially remained high during the first days of infection by the pathogen. However, both expression levels and enzyme activity decreased as time progressed (Figure 2c and Figure 3b). A similar behavior was obtained in the study by Luján-Hidalgo et al. [29] in coffee, where peroxidase activity decreased after infection, which was associated with a beneficial response to avoid infection and was related to its responsibility in reinforcing the cell wall of the plant. In the study by Varghese et al. [30], they observed the same trend in ginger, where chitosan did not present an increase in peroxidase activity; however, the accumulation of lignin, one of the polymers associated with cell wall strengthening, was observed.

In addition, there are specific peroxidases that face pathogen invasion and can be induced by chitosan, including SOD, which is the first enzyme in the ROS detoxification process, reacting with O_2_^−2^ superoxide radicals to produce H_2_O_2_ [31]. In this work, this group of enzymes had similar kinetics to peroxidases; they remained stable during the first evaluation times, and at 72 h, there was a decrease in activity (Figure 2d). In the same sense, APX, which catalyzes the reaction to convert H_2_O_2_ into H_2_O and O_2_, showed a similar response, although at 12 h, it had high activity, which decreased as time passed. For CAT, a tetrameric enzyme for ROS detoxification that converts H_2_O_2_ to H_2_O and is the first enzyme to act in the presence of oxidative stress [31], the results showed increased gene expression throughout the times evaluated. These results agree with those reported by López-Velázquez et al. [14], where CAT activity increased with the application of chitosan, and SOD remained without significant changes. The study conducted by Zhang et al. [32] reported that chitosan had an important effect on the activation of the antioxidant system of plants since enzyme activity was favored, suggesting that chitosan regulates in favor of the plant defense system and its metabolic activation for metabolite synthesis.

However, the decrease in peroxidase activity may be related to redirection to other defense mechanisms, e.g., the induction of PCD and oxidative burst, and effective defense mechanisms against biotrophic pathogens such as coffee rust [25]. Biotrophic fungi, as part of their basic pathogenesis mechanisms, present fungal effectors that inhibit PCD induction, including inducers of antioxidant mechanisms aimed at cutting the role as signal molecules of ROS and defensive processes associated with their proliferation [33]. Although not explored in this work, it raises the prospect for future work to explore the role of cell wall strengthening, PCD induction, and ROS accumulation in resistance to rust infection in the coffee plant.

On the other hand, phenolic compounds contribute to maintaining overall plant health by regulating a wide range of functions, such as protection against pathogens, chelation of heavy metals, and prevention against oxidative stress. They provide mechanical strength to the cell, forming lignin and suberin, two important constituents of the cell wall [34]. In this study, the content of phenolic compounds had a better effect compared to plants infected with *H. vastatrix* (H), which, unlike the other treatments evaluated, decreased. In the work carried out by Guo et al. [35], it was observed that chitosan induced the accumulation of total phenolic compounds; however, the content of these compounds was lower compared to the control, as in this case. Peian et al. [35] reported an increase in the content of total phenolic compounds in strawberry and grape berries that were treated with chitosan when exposed to *Botrytis cinerea*.

Phenolic compounds are generated from the shikimic acid pathway and are precursors of phenylpropanoids through the activity of the PAL enzyme, where phenylalanine undergoes deamination and becomes a precursor of flavonoids, lignin, and other defense compounds [36]. In this work, *PAL* gene expression was monitored with respect to infected plants (H); the expression increased 5-fold at 12 h and almost 45-fold at 24 h, reaching its maximum expression at 48 h, which was related to what was observed in the content of total phenolic compounds. Comparing the results with Godana et al. [36], PAL activity was favored at 2 days when the grapes were treated with chitosan and inoculated with *Pichia anomala*, after which the activity decreased, suggesting that there would be a production of total phenolic compounds. Another important aspect is the relationship with polyphenol oxidase activity, which is an enzyme that catalyzes the oxidation of phenolic compounds into highly reactive quinones, which help plants defend themselves against pathogens [22]. The results showed that enzyme activity was favored with chitosan application (QH) compared to infected plants (H), and a directly proportional relationship was found with the accumulation of total phenolic compounds. Chitosan also plays a fundamental role in the induction of enzymatic activity, as observed in the work of Liu et al. [22], where it increased the levels of this activity in potatoes. In addition, it also plays a role in the thickening of the plant cell wall, which serves as a physical barrier to prevent colonization by pathogens [28]. A similar effect was found in the study by Ahmad et al. [37], where PPO activity was induced by chitosan and was associated with a beneficial response in the control of citrus diseases, and the amount of total phenolic compounds was increased.

In this study, it was observed that food-grade chitosan was able to induce effective protection against *H. vastatrix* infection in coffee of an intensity equivalent to that of practical-grade chitosan. Chitosan was able to induce a differentiated defense response with respect to rust-infected plants. It significantly increased the activity of β-1,3 glucanases. Antioxidant enzyme activity was also different from that observed in rust-infected coffee plants. However, it was observed that most of them presented lower activity than that observed in infected plants, which could be related to the role acquired by ROS in triggering an effective defense response against coffee rust or by directing these molecules to the strengthening of the cell wall. It was able to induce the production and accumulation of phytoalexins by inducing the expression of *PAL* and *PPO* genes and the content of total phenolic compounds. However, the nature of these compounds and their role in the effective defense mechanism against rust remains to be defined. Finally, chitosan was able to induce *NPR1* gene expression, indicating its role in the SA signaling pathway and in the dissemination of an effective systemic defense response. From these results, a better understanding of the effective defense mechanisms induced by chitosan in the protection against *H. vastatrix* infection in coffee plants was obtained. It is important to consider the evaluation of the role of chitosan in cell wall strengthening, PCD induction, and ROS accumulation in resistance to rust infection in coffee plants.

## 4. Materials and Methods

### 4.1. Plant Material

Coffee fruits (*Coffea arabica* var. Typica) were collected in the locality of Tlajomulco de Zúñiga, Jalisco, Mexico. The fruits were pulped, and the seeds were recovered. The seeds were washed with water and disinfected with 3% (*v*/*v*) sodium hypochlorite for 20 min. They were then placed in a drying oven (Drying, DH6-9145A, Xi’an, China) at 30 °C ± 2 °C for 48 h. Subsequently, seeds were germinated in a mixture of peat moss with vermiculite (5:2) in germination trays, which were placed in an incubation room at 26 °C ± 2 °C, with a photoperiod of 16 h light/8 h dark. Plants were fertilized every week with macro- and micronutrients and were used for experiments at the age of 6 months [38].

### 4.2. Biological Material

Uredinospores of the phytopathogenic fungus *H. vastatrix* were collected from plants with rust symptoms in the municipality of Ejutla, Jalisco, Mexico. The spores were recovered and resuspended in distilled water with 0.05% (*v*/*v*) tween 20 (Sigma-Aldrich, Poole, UK). The solution was adjusted to a concentration of 1 × 10^5^ uredinospores/mL and sprayed on the underside of the leaves. The plants were watered, covered with a polyethylene bag, and kept in darkness for 72 h in an incubation room at 26 °C ± 2 °C [39]. After 72 h, the plants were kept in a 16/8 light/dark photoperiod at 26 °C ± 2 °C.

### 4.3. High-Density Chitosan Solution

A 1% (*w*/*v*) solution of high-density food-grade chitosan (America Alimentos, Guadalajara, Jalisco, Mexico) was prepared. The chitosan was dissolved in a 0.4 M acetic acid solution (Fermont, Guadalajara, Jalisco, Mexico) with 2 M sodium acetate (Sigma-Aldrich, St. Louis, MO, USA). Finally, a 0.01% and 0.05% dilution of chitosan was performed with distilled water.

### 4.4. Biological Effectiveness Test with Food-Grade Chitosan

The plants were distributed in four treatments with ten plants each: control plants treated with distilled water (C), plants inoculated with *H. vastatrix* (H), plants treated with 0.01% food-grade chitosan and inoculated with the pathogen (Q1), and plants treated with 0.05% food grade chitosan and inoculated with the pathogen (Q5). Treatments C and H were sprayed with an atomizer with 3 mL of distilled water, and treatments Q1 and Q5 with 3 mL of food-grade chitosan on all leaves. Seven days after water and chitosan application, plants were inoculated with 3 mL of *H. vastatrix* urediniospore suspension. The plants were kept in an incubation room at 26 °C ± 2 °C and were monitored for 90 days (40, 60, and 90 days after inoculation) to observe the development of the disease, taking as a reference Technical Data Sheet 40, Coffee Rust, SENASICA [40]. The experiment was carried out in December 2021 and March 2022.

#### 4.4.1. Disease Incidence

The plants were observed at 40, 60, and 90 days after inoculation, and disease incidence was calculated according to Equation (1). Each treatment had 10 plants [41].
(1)$$\mathrm{Disease}\;\mathrm{incidence}\;(\%)=\frac{NIP}{TIP}\;\left(100\right),$$
where *NIP* corresponds to the number of infected plants, and *TIP* corresponds to the total number of infected plants assessed.

#### 4.4.2. Area under the Disease Progress Curve (AUDPC)

Data were collected on the severity of infection at 40, 60, and 90 days after inoculation, and the area under the disease progress curve was determined according to Equation (2) [42].
(2)$$\mathrm{AUDPC}=\sum_{i=1}^{n}\left[\frac{y_{i}+y_{i+1}}{2}](x_{i}-x_{i-1}\right)$$
where AUDPC is the area under disease progress curve, *y_i_* is the percentage of visible infectes area (*y_i_*/100) at the *i*-th observation, *x_i_*—day of the *i*-th observation, and *n* the total number of observations.

#### 4.4.3. Observation of the Presence of Pathogen in the Plant Tissue

Leaf samples were taken 90 days after inoculation with the pathogen. Leaves were washed with distilled water to remove particles prior to staining and placed in 50 mL conical tubes. The samples were bleached with 1.7 M KOH at 50 °C for 30 min. Subsequently, they were washed with sterile distilled water twice and 0.5 M hydrogen peroxide at 50 °C for 15 min. Finally, they were stained with 0. 05% (*w*/*v*) in lactoglycerol (lactic acid: glycerol: water; 1:1:1:1, *v*/*v*) at 50 °C for 30 min. Excess dye was removed with lactoglycerol and observed under an optical microscope (Olympus model BH-2, Tokyo, Japan) [43].

### 4.5. Enzymatic Activity Evaluation Related to the Pathogenesis and Accumulation of Phytoalexins

Leaves were sampled before inoculation (0 h) and 12, 24, 48, and 72 h after inoculation with the pathogen. Leaves were frozen in liquid nitrogen and stored at −80 °C until processing. In total, 200 mg of fresh tissue was weighed and macerated in liquid nitrogen in a mortar with 1% polyvinylpyrrolidone. For the determination of enzymes with superoxide dismutase and ascorbate peroxidase activity, the pulverized tissue was mixed with potassium phosphate buffer pH 7.8, 0.1 mM EDTA, and 10 mM ascorbic acid, and then centrifuged at 13,000× *g* for 25 min at 4 °C. For the determination of β-1,3 glucanase, peroxidase, and polyphenol oxidase activity, the tissue was resuspended in potassium phosphate buffer pH 7.0 and 0.1 mM EDTA, then centrifuged at 13,000× *g* for 25 min at 4 °C [8]. To determine phytoalexins, 0.1 g of tissue was resuspended in 500 µL of 80% methanol [11]. On each sampling day, 3 plants were taken for the corresponding evaluation, and each test was performed in triplicate. For all assays, 200 uL of sample were used and read in a 96-well microplate.

#### 4.5.1. β-1,3 Glucanase Activity

β-1,3 glucanase activity was determined with the colorimetric method for the detection of reducing sugars DNS (3,5-Dinitrosalicylic acid), quantification was determined with a calibration curve with glucose, and activity was reported in nkat per g of total protein (nkat·g^−^^1^·g^−^^1^ PT) [11]. The read was carried out in a spectrophotometer (Thermo Fisher Scientific, Multiskan Go FI-01620, Vantaa, Finland).

#### 4.5.2. Peroxidase Activity

Perioxidase activity determination was carried out with the guaiacol oxidation method. The variation of one absorbance unit per minute was defined as one unit of peroxidase activity (1 AU) and was expressed per gram of total protein (UA·g^−^^1^ PT) [11]. The read was carried out in a spectrophotometer (Thermo Fisher Scientific, Multiskan Go FI-01620, Vantaa, Finland).

#### 4.5.3. Superoxide Dismutase Activity

Superoxide dismutase activity was evaluated by the ability of the SOD enzyme to inhibit the photoreduction of nitro blue tetrazolium (NBT) prepared in an incubation medium composed of 50 mM potassium phosphate, pH 7.8, 14 mM methionine, 0.1 µM EDTA, 75 µM NBT, 2 µM riboflavin, and plant extract, then incubated for 7 min with a 30 W fluorescent lamp and read at 560 nm in a spectrophotometer (Thermo Fisher Scientific, Multiskan Go FI-01620, Vantaa, Finland). One unit of SOD is considered the amount of enzyme capable of being inhibited by 50% NBT photoreduction under assay conditions. The units were reported in U^−^^1^·min^−^^1^·mg^−^^1^·TP [8].

#### 4.5.4. Ascorbate Peroxidase Activity

Twenty µL of the plant extract were added to 200 µL of 100 mM potassium phosphate solution at pH 7.0, 0.5 mM ascorbic acid, 0.1 mM hydrogen peroxide, and water. The activity was determined by consuming H_2_O_2_ at 290 nm for 1 min at 25 °C in a spectrophotometer (Thermo Fisher Scientific, Multiskan Go FI-01620, Vantaa, Finland). The molar extinction coefficient of 1.4 mM·cm^−^^1^ was used to calculate APX activity, and the units were reported in µM^−^^1^·min^−^^1^·mg^−^^1^·TP [8].

#### 4.5.5. Polyphenol Oxidase Activity

Twenty µL of the plant extract were added to 200 µL of 70 mM potassium phosphate solution at pH 7.0 and 20 mM catechol. Then, it was incubated at 30 °C for 10 min. Absorbance was measured at 410 nm in a spectrophotometer (Thermo Fisher Scientific, Multiskan Go FI-01620, Vantaa, Finland). The molar extinction coefficient of 1.235 mM·cm^−^^1^ was used to calculate the PPO activity, and the units were reported in mM^−^^1^·min^−^^1^·mg^−^^1^·TP [8].

#### 4.5.6. Protein Quantification

Protein quantification was performed using the Bradford method described by Kruger [44], using bovine albumin for the calibration curve. The read was carried out in a spectrophotometer (Thermo Fisher Scientific, Multiskan Go FI-01620, Vantaa, Finland).

#### 4.5.7. Phytoalexin Accumulation

Phytoalexin accumulation was performed by quantification of total phenolic compounds by Folin–Ciocalteu’s method and a catechol curve was used at known concentrations. The accumulation of phenolic compounds was reported in nM catecol·mg^−^^1^ FW [11]. The read was carried out in a spectrophotometer (Thermo Fisher Scientific, Multiskan Go FI-01620, Vantaa, Finland).

### 4.6. Quantitative Analysis of Defense-Related Gene Expression by Real-Time PCR

The plants were distributed in four treatments: control plants treated with distilled water (C), plants inoculated with *H. vastatrix* (H), plants treated with 0.05% high-density chitosan (Q), and plants treated with 0.05% high-density chitosan and inoculated with the pathogen (QH). Three mL of distilled water were sprayed in treatments C and H, and 3 mL of high-density chitosan in treatments Q and QH. Seven days after the application of water or chitosan, 3 mL were inoculated with the suspension of urediniospores of *H. vastatrix*. Plants were kept in an incubation room at 26 °C ± 2 °C, and leaf samples were taken at 12, 24, and 48 h after inoculation with the pathogen [9]. On each sampling day, 3 plants were taken for the corresponding evaluation, and each test was performed in duplicate.

#### 4.6.1. Primers Design

Primers coding for the genes *peroxidase* (*POX*), *catalase* (*CAT*), *ß-1,3-glucanase* (*GLU*), and *phenylalanine ammonia lyase* (*PAL*) were designed. The genomic sequences were obtained from the Phytozome repository. Subsequently, the OligoAnalyzer Tool from IDT and Primer-BLAST from NCBI were used. Additionally, *glyceraldehyde-3-phosphate dehydrogenase* (*GADPH*), *actin 8*, and *14-3-3* primers were designed as constitutive controls. Table 2 shows the characteristics of each pair of oligonucleotides. The stability of the primers was determined according to the method of Vandesompele et al. [45].

#### 4.6.2. RNA Extraction and cDNA Synthesis

The TRIzol extraction protocol (Invitrogen, Carlsbad, CA, USA) was used with some modifications) [46]. In total, 200 mg of sample were macerated with liquid nitrogen, and 1 mL of TRIzol was added for 5 min. The samples were centrifuged at 11,000× *g* for 15 min at 4 °C, the supernatant was removed, and 500 µL chloroform (Sigma-Aldrich, Dorset, UK) were added, shaken for 20 min at −20 °C, then centrifuged at 11,000× *g* for 15 min at 4 °C. the supernatant was removed, 500 µL chloroform (Sigma-Aldrich, Dorset, UK). Subsequently, the supernatant was recovered, and 500 µL of isopropanol (Sigma-Aldrich, Dorset, UK) were added, left to stand for 2 h at −80 °C, and centrifuged at 11,000× *g* for 15 min at 4 °C, after which the supernatant was removed. Three washes were performed with 75% ethanol (Sigma-Aldrich, St. Louis, MO, USA), and the pellet was dried and resuspended in 20 µL of RNAse-free water. RNA was quantified at 260 nm and migrated on 0.8% agarose gel. For cDNA synthesis, the Maxima H Minus First Strand cDNA Synthesis kit (Thermo Fisher Scientific Baltics UAB, Vilnius, Lithuania) was used. Synthesis was performed according to the manufacturer’s instructions, starting from 100 ng of total RNA.

#### 4.6.3. RT qPCR

RT qPCR analyses were performed on a Step-One Real-Time PCR System thermal cycler (Applied Biosystems, Life Technologies, San Francisco, CA, USA) using SYBR Green as the detection system. The reaction conditions were 2 min at 50 °C, 10 min at 60 °C, followed by 40 cycles of 15 s at 95 °C, and 1 min at 60 °C, ending the cycle with 15 s at 95 °C. The reaction was carried out with 50 ng of cDNA, 0.2 µL of each primer, and 7.5 µL of SYBR green PCR Master Mix (Invitrogen, St. Louis, MO, USA) and RNAse-free water. For the evaluation of each gene, three biological samples with two technical replicates were performed. Relative quantification (RQ) was performed using the ddCT method [47].
RQ = 2^−ddCT^; ddCT = [(CT gene of interest − CT constitutive gene) treated plant − (CT gene of interest − CT constitutive gene) untreated control plant].(3)

### 4.7. Statistical Analysis

One-way analysis of variance (ANOVA) was performed using Minitab 19.2020.2.0 software, followed by a separation of means through the LSD test, with a significance level of 95%.

## 5. Conclusions

The application of 0.05% high-density food-grade chitosan significantly reduced the severity of *H. vastatrix* infection by increasing the activity and gene expression of *β-1,3 glucanases*. In addition, a differentiated response was found in enzymes related to the antioxidant system of plants. Additionally, a correlation was observed between the activities of the enzyme polyphenol oxidase and the production of total phenolic compounds that allowed an effective defense response in coffee plants.

## Figures and Tables

**Figure 1 ijms-24-16165-f001:**
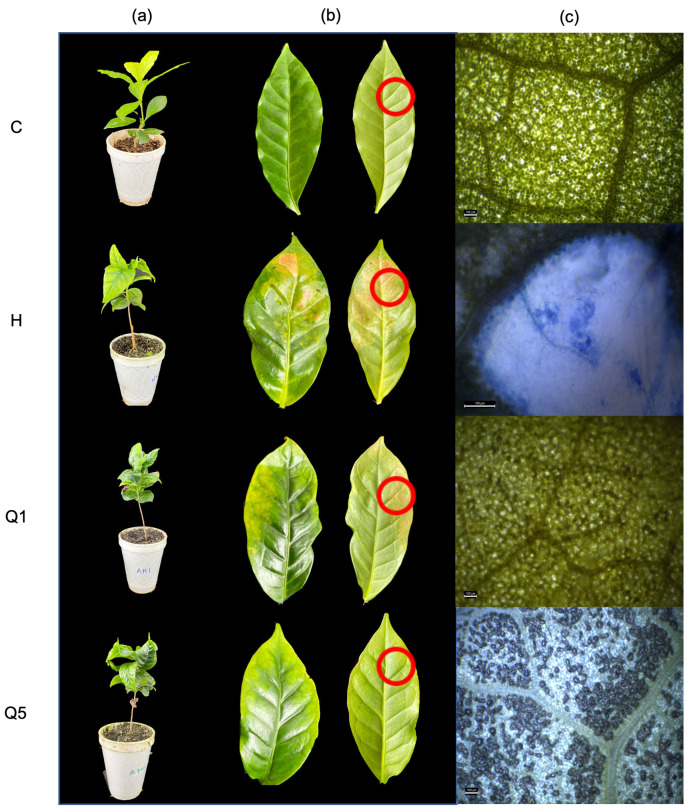
Biological effectiveness test 90 days after inoculation. (**a**) Biological effectiveness test, (**b**) disease leaves, and (**c**) presence of pathogen. (C) Plants without treatment, (H) plants inoculated with *Hemileia vastatrix*, (Q1) plants sprayed with 0.01% high-density chitosan and inoculated with *H. vastatrix*, and (Q5) plants sprayed with 0.05% high-density chitosan and inoculated with *H. vastatrix*. The red circles indicate the area of the leaf where the pathogen was observed.

**Figure 2 ijms-24-16165-f002:**
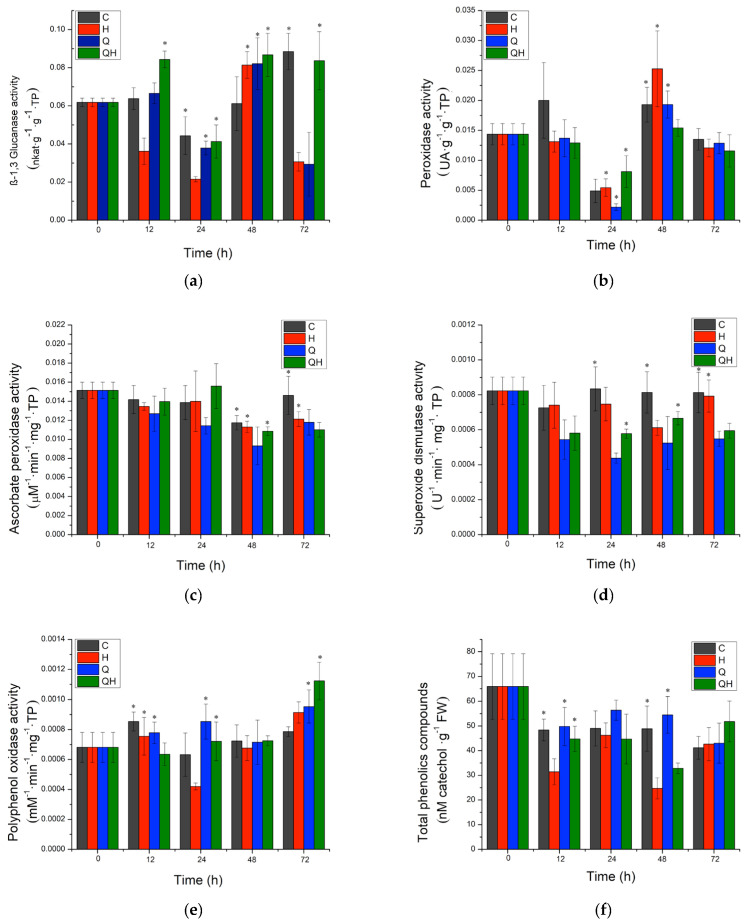
PR protein activity of (**a**) β-1,3 glucanase, (**b**) peroxidase, (**c**) ascorbate peroxidase, (**d**) superoxide dismutase, and (**e**) polyphenol oxidase. Phytoalexin accumulation of (**f**) total phenolics compounds. (C) Plants without treatment, (H) plants inoculated with *Hemileia vastatrix*, (Q) plants sprayed with 0.05% food chitosan, and (QH) plants sprayed with 0.05% food chitosan and inoculated with *H. vastatrix*. Values presented are means with standard deviation. Values with asterisks show statistically significant differences at the 95% LSD confidence level (*p* < 0.05).

**Figure 3 ijms-24-16165-f003:**
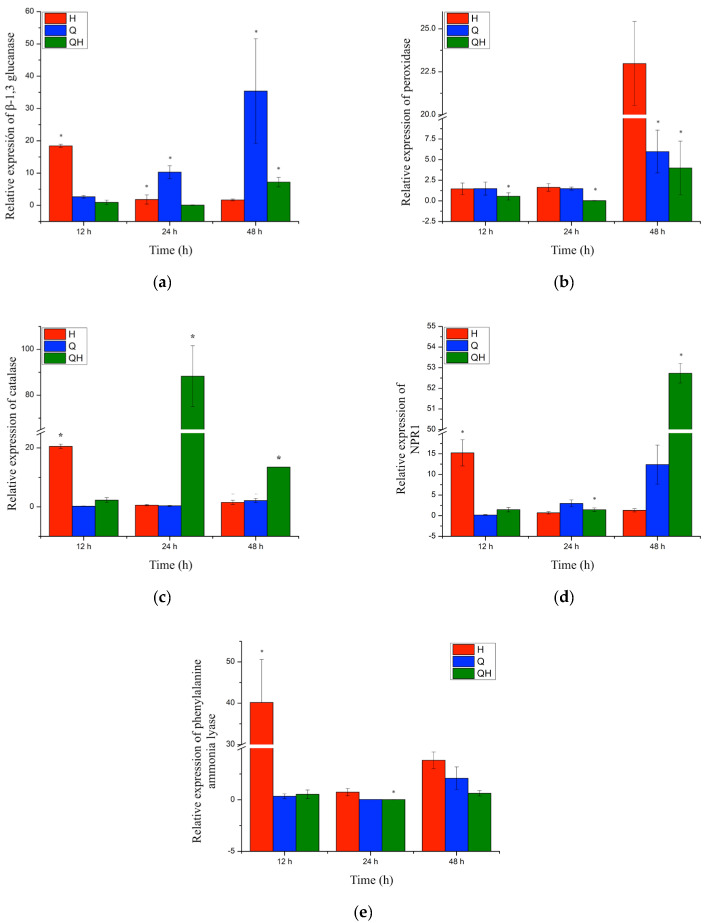
Relative expression of genes (**a**) *β-1,3 glucanase*, (**b**) *peroxidase*, (**c**) *catalase*, (**d**) *NPR1*, (**e**) and *phenilalanine ammonia lyase*. (H) Plants inoculated with *Hemileia vastatrix*, (Q) plants sprayed with 0.05% food chitosan, and (QH) plants sprayed with 0.05% food chitosan and inoculated with *H. vastatrix*. Values presented are means with standard deviation. Values with asterisks show statistically significant differences at the 95% LSD confidence level (*p* < 0.05). Plants without treatment were used as a baseline.

**Table 1 ijms-24-16165-t001:** Disease parameters evaluated.

Treatment	Disease Incidence (%)	AUDPC (au)
C	0	0.0 ± 0.0 c
H	100	102 ± 14.3 a
Q1	100	40.0 ± 20.6 b
Q5	100	25.0 ± 11.66 b

The treatments correspond to: (C) plants without treatment, (H) plants inoculated with *Hemileia vastatrix*, (Q1) plants sprayed with 0.01% high-density chitosan and inoculated with *H. vastatrix*, and (Q5) plants sprayed with 0.05% high-density chitosan and inoculated with *H. vastatrix*. Values presented are means with standard deviation. Values with different letters show statistically significant differences at the 95% LSD confidence level (*p* < 0.05). au: Arbitrary units.

**Table 2 ijms-24-16165-t002:** Oligonucleotides.

Genes	Oligonucleotide Sequences (5′-3′)	Size (pb)	Accession Number
*Peroxidase (POX)*	FW: GTGGATGCGGAGTACCTGAA RV: AACCGTTTGGACCTCCAGTT	153	evm.TU.Scaffold_618.242
*Catalase (CAT)*	FW: GATGCACCCAATTCCTTCTGC RV: CCCAGCGACAGATAAAGCG	140	evm.TU.Scaffold_2256.195
*ß-1,3 glucanase*	FW: GGGTGACCCTACAAAAGCCA RV: GGCCTGGAGGAAAGGTTCAT	121	evm.TU.Scaffold_517.1
*Phenylalanine ammonia* *lyase (PAL)*	FW: CTTGTGAGGGGAGAGTTGGG RV: GGTAGGTGGCTCTTGTCAGC	112	evm.TU.Scaffold_618.1114
*NPR1*	FW: TTGGTTATGAGGCTGCTGCT RV: GGCTTTAGATGCTGCAAGGC	138	evm.model.Scaffold_523.152
*GAPDH*	FW: CCCTTGGGGTGAAACTGGAG RV: AACATGGGTGCATCCTTGCT	138	evm.model.Scaffold_608.230
*Actin 8*	FW: ATTAGCTCGACAAGACGCCC RV: CTCACGTTCCATGTGTTGCG	142	evm.model.Scaffold_315.601
*14-3-3*	FW: GCTGAGTTCAAAACTGGGGCTG RV: ATTGGGTGTGTTGGAGCCAG	110	evm.model.Scaffold_2421.275

## Data Availability

The data presented in this study are available on request from the corresponding author.

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
