# Peer review of "High-Density Chitosan Induces a Biochemical and Molecular Response in Coffea arabica during Infection with Hemileia vastatrix"

_ijms, 2023, doi:10.3390/ijms242216165_

Round 1

Reviewer 1 Report

Comments and Suggestions for Authors

Dear Authors,

After reading the paper, it is clear that the authors performed analyze in order to obtain detailed research results concerning on the influence of high-density chitosan on response in Coffea arabica and defense of plant against Hemileia vasta trix. In general, the manuscript is easy to read, the structure of the work is clear, however, chapter “discussion” is large and extensive and cover many paragraph concerned on other species and other disease. Some of them are not necessary. After reading the paper, it is clear that the authors performed experiments on plants of Typica variety of Coffea Arabica, which fruits were collected from Tlajomulco de Zúñiga, Jalisco, Mexico. The fruits were washed (washed method, I guess) and after 48 hour fermentation beans (seeds) were dried for 48 h. at 30oC. It is clear, the beans was healthy seed material, which was used for germination in order to obtain plants after 6 months. However, why plant material with rust symptoms (Hemileia vasta trix) was used from other place; the municipality of Ejutla, Jalisco, Mexico?

I present my comments below:

Results are adequate to the methods used in this paper. All results presented of Figures 2(a-f) and 3(a-e) are described detailed as are presented on figures. However, according to the Anova test standard deviation shows not significant differences frequently. It is the reason that you should indicate tendency only, not the influence. On the other hand, all figures 2(a-f) are too small and colored lines as well as standard deviation overlap each other, that is difficult state significant differences. Similar on figure 3 (a-e) standard deviation is cover by color bar, and frequently is not visible because of figures size, as well as description of axis are to small fonts.

In addition, the measurement were realized 4 times, at 12, 24, 48 and 72 hours, and all points was connected with strait lines. Measuring points presented on figures should be performed as points or bars (like on fig 3) including confidence interval only. It is well known, that two points representing measuring data did not connect with strait line, because from mathematical point of view two points may be connect with a line of any shape, not strait line only.

1. Lines 227-241 – both paragraphs  move to the chapter “Introduction”

2. Lines 460 – 461 – If you show formula, you should use symbols, and it should looks as follow: Di=n/nT  x100% or    and all symbols you used can be listed under formula.

3. Lines 465-468 - Formula 2 should looks similar to formula 1. If you not change keep attention on  all indexes, because are not subscripts (formula 2 and lines 466-468).

4. Because statistical methods you used did not indicate significant differences, try to use other statistical method e.g. PCA (look "How to identify roast defects in coffee beans based on the volatile compound profile" or other.  

Generally, after reading the work seems complete; however, chapter "discussion" cover some paragraphs weakly related to the results obtained in presented paper and some of them (including references) may be missed or moved to the chapter “Introduction”. As I said, there are too many number of literature cited, maybe some of them are not necessary, especially concerning on other species of plant and other disease, however decision to erasing is up to you.

Comments on the Quality of English Language

The manuscript is written in understandable language for reader.

Author Response

Response to Reviewer 1 Comments

1. Summary

Thank you very much for taking the time to review this manuscript. Please find the detailed responses below and the corresponding revisions and corrections highlighted changes in the re-submitted files.

2. Questions for General Evaluation

Reviewer’s Evaluation

Response and Revisions

Does the introduction provide sufficient background and include all relevant references?

Yes

Are all the cited references relevant to the research?

Can be improved

Some non-essential references were selected and removed.

Is the research design appropriate?

Yes

Are the methods adequately described?

Yes

Are the results clearly presented?

Can be improved

The graphs were modified according to the reviewer's comments.

Are the conclusions supported by the results?

Yes

3. Point-by-point response to Comments and Suggestions for Authors

Comments 1: After reading the paper, it is clear that the authors performed analyze in order to obtain detailed research results concerning on the influence of high-density chitosan on response in Coffea arabica and defense of plant against Hemileia vastatrix. In general, the manuscript is easy to read, the structure of the work is clear, however, chapter “discussion” is large and extensive and cover many paragraph concerned on other species and other disease. Some of them are not necessary. After reading the paper, it is clear that the authors performed experiments on plants of Typica variety of Coffea Arabica, which fruits were collected from Tlajomulco de Zúñiga, Jalisco, Mexico. The fruits were washed (washed method, I guess) and after 48 hour fermentation beans (seeds) were dried for 48 h. at 30oC. It is clear, the beans was healthy seed material, which was used for germination in order to obtain plants after 6 months. However, why plant material with rust symptoms (Hemileia vastatrix) was used from other place; the municipality of Ejutla, Jalisco, Mexico?

Response 1: Typica coffee is grown in Tlajomulco de Zuñiga in an isolated site and is not intensively cultivated. Therefore, it is not infected with coffee rust. This assures us of a seed stock without obvious phytosanitary problems. On the other hand, in the municipality of Ejutla, Jalisco. The crop where the collection of coffee rust was carried out, the plantations do present symptoms of the disease. The conditions where this crop is located have a particular microclimate that allows the development of both the coffee crop and the symptoms of rust in a cyclical manner.

Comments 2: Results are adequate to the methods used in this paper. All results presented of Figures 2(a-f) and 3(a-e) are described detailed as are presented on figures. However, according to the Anova test standard deviation shows not significant differences frequently. It is the reason that you should indicate tendency only, not the influence. On the other hand, all figures 2(a-f) are too small and colored lines as well as standard deviation overlap each other, that is difficult state significant differences. Similar on figure 3 (a-e) standard deviation is cover by color bar, and frequently is not visible because of figures size, as well as description of axis are to small fonts.

In addition, the measurement were realized 4 times, at 12, 24, 48 and 72 hours, and all points was connected with strait lines. Measuring points presented on figures should be performed as points or bars (like on fig 3) including confidence interval only. It is well known, that two points representing measuring data did not connect with strait line, because from mathematical point of view two points may be connect with a line of any shape, not strait line only.

Response 2: Agree. The graphs were modified according to the reviewer's comments.

Figure 2. Page 6.

Figure 3. Page 7-8.

Comments 3: Lines 227-241 – both paragraphs move to the chapter “Introduction”

Response 3: Agree. Both paragraphs were incorporated into the “introduction” and repetitive information was eliminated.

Page 2, Lines 45-49

Comments 4: Lines 460 – 461 – If you show formula, you should use symbols, and it should looks as follow: Di=n/nT  x100% or    and all symbols you used can be listed under formula.

Response 4: Agree. We accordingly, modified the formula.

Disease incidence (%)

Where: NIP corresponds to Number of infected plants and TIP corresponds to Total number infected of plants assessed

Page 12, Lines 48-50.

Comments 5: Results are adequate to the methods used in this paper. All results presented of Figures 2(a-f) and 3(a-e) are described detailed as are presented on figures. However, according to the Anova test standard deviation shows not significant differences frequently. It is the reason that you should indicate tendency only, not the influence. On the other hand, all figures 2(a-f) are too small and colored lines as well as standard deviation overlap each other, that is difficult state significant differences. Similar on figure 3 (a-e) standard deviation is cover by color bar, and frequently is not visible because of figures size, as well as description of axis are to small fonts.

In addition, the measurement were realized 4 times, at 12, 24, 48 and 72 hours, and all points was connected with strait lines. Measuring points presented on figures should be performed as points or bars (like on fig 3) including confidence interval only. It is well known, that two points representing measuring data did not connect with strait line, because from mathematical point of view two points may be connect with a line of any shape, not strait line only.

Response 5: Agree. We accordingly, modified the formula.

AUDPC=

The equation was put into a more understandable format.

Page 12, lines 455-456

Comments 6: Because statistical methods you used did not indicate significant differences, try to use other statistical method e.g. PCA (look "How to identify roast defects in coffee beans based on the volatile compound profile" or other.

Response 6: Other statistical analyzes were carried out, such as the comparison of means using the Tukey test, although it presented a lower significant difference than the selected statistical method. The proposed principal components analysis (PCA) was also carried out. However, it was not possible to group the variables, they remained independent. Therefore, it was considered that the selected statistical analysis was appropriate to explain the results.

Comments 7: Generally, after reading the work seems complete; however, chapter "discussion" cover some paragraphs weakly related to the results obtained in presented paper and some of them (including references) may be missed or moved to the chapter “Introduction”. As I said, there are too many number of literature cited, maybe some of them are not necessary, especially concerning on other species of plant and other disease, however decision to erasing is up to you.

Response 7:  Agree. We accordingly, some non-essential references were selected and removed.

Page 16-18, lines 618-743.

4. Response to Comments on the Quality of English Language

Point 1: The manuscript is written in understandable language for reader.

Response 1: A final revision of the manuscript will be carried out through the language editing by MDPI.

5. Additional clarifications

Reviewer 2 Report

Comments and Suggestions for Authors

The authors reported on the biochemical and molecular response of coffee plants to Hemileia vastatrix infection, which causes coffee rust disease. High-density chitosan was chosen to combat the disease and improve coffee production.

The manuscript is well written and organized. I recommend the publication of this manuscript on Int.       J. Mol. Sci. upon the following conditions are well addressed.

[1] Why the authors did not use higher concentration of chitosan (>0.05%) in the experiments?

[2] In Table 1, there are no clear descriptions for the notes “a”, “b” and “c” under AUDPC. Also, the unit of AUDPC is missing.

[3] In Figure 1c, the authors better choose the similar regions of the leaves for observing the presence of pathogen for fair comparison.

[4] Would the use of chitosan change the favor and taste of the coffee?

Comments on the Quality of English Language

Minor editing of English language required

Author Response

Response to Reviewer 2 Comments

1. Summary

Thank you very much for taking the time to review this manuscript. Please find the detailed responses below and the corresponding revisions and corrections highlighted changes in the re-submitted files.

2. Questions for General Evaluation

Reviewer’s Evaluation

Response and Revisions

Does the introduction provide sufficient background and include all relevant references?

Yes

Are all the cited references relevant to the research?

Yes

Is the research design appropriate?

Yes

Are the methods adequately described?

Yes

Are the results clearly presented?

Yes

Are the conclusions supported by the results?

Yes

3. Point-by-point response to Comments and Suggestions for Authors

Comments 1: Why the authors did not use higher concentration of chitosan (>0.05%) in the experiments?

Response 1: Previous studies have evaluated different concentrations of chitosan in the control of coffee rust.

·        López-Velázquez, J.C.; Haro-González, J.N.; García-Morales, S.; Espinosa-Andrews, H.; Navarro-López, D.E.; Montero-Cortés, M.I.; Qui-Zapata, J.A. (2021) Evaluation of the Physicochemical Properties of Chitosans in Inducing the Defense Response of Coffea arabica against the Fungus Hemileia vastatrix. Polymers 13, 1940. https://doi.org/10.3390/polym13121940

·        López-Velázquez JC, García-Morales S, Espinosa-Andrews H, Montero-Cortés MI, Qui-Zapata JA (2022) Control de la roya del café utilizando moléculas de quitosano de diferentes orígenes. Memorias del XLIII Encuentro Nacional de la AMIDIQ, 23 al 26 de agosto de 2022. Puerto Vallarta, Jal. Avances en ingeniería química, Vol. 2. No. 1, BIO 151-BIO 156. (ISSN 2683-2925).

https://www.researchgate.net/profile/Joaquin-Qui/publication/370682176_CONTROL_DE_LA_ROYA_DEL_CAFE_UTILIZANDO_MOLECULAS_DE_QUITOSANO_DE_DIFERENTES_ORIGENES/links/645d1f63f43b8a29ba44e7e9/CONTROL-DE-LA-ROYA-DEL-CAFE-UTILIZANDO-MOLECULAS-DE-QUITOSANO-DE-DIFERENTES-ORIGENES.pdf

It has been found that within this range of concentrations there is control of coffee rust. In this study both concentrations are taken up and the best concentration in the biological effectiveness test was 0.05%.

Comments 2: In Table 1, there are no clear descriptions for the notes “a”, “b” and “c” under AUDPC. Also, the unit of AUDPC is missing.

Response 2: The letters are not notes, the letters show statistically significant differences at the 95% LSD confidence level (p < 0.05). The letters are shown as normal text.

AUDPC values were obtained from a previously reported symptom scale, which indicates the severity of the disease, and includes the time interval in the evaluated days mentioned in the methodology: 40, 60 and 90 days after infection. The units are arbitrary (au).

The reference where this measurement is clarified is the following:

https://doi.org/10.1094/PHYTO.1997.87.5.506

Page 3, lines 100-101.

Page 3, Table 1, lines 111-116.

Comments 3: In Figure 1c, the authors better choose the similar regions of the leaves for observing the presence of pathogen for fair comparison.

Response 3: The disease is not distributed homogeneously, the sites where symptoms caused by the pathogen were observed were chosen, the evaluated areas are identified with a red circle in Figure 1b.

Page 4, Figure 1. Line 122.

Comments 4: Would the use of chitosan change the favor and taste of the coffee?

Response 4: In this study, parameters related to growth, plant development and the effect of flavor on coffee were not evaluated. The work aimed to evaluate changes related to the induction of a plant defense at the biochemical and molecular level, in which it has the function as a plant defense inducer through a phenomenon called "priming", where molecules are used that act as response modifiers that can lead to a more intense, faster, earlier or more sensitive defense response compared to plants that are exposed to the same stress condition, as mentioned by Tugizimana et al, 2018 (https://doi.org/10.3390/ijms19061759 ).

In addition, because of the age of plants used, fruits cannot be obtained at that phenological stage.

4. Response to Comments on the Quality of English Language

Point 1: Minor editing of English language required.

Response 1: A final revision of the manuscript will be carried out through the service language editing by MDPI.

5. Additional clarifications
